# "All Eyes On Me": User Perspective on Self-View in Work From Home Video Conferencing

Anon*
Anon

## ABSTRACT

Online video conferencing is not a new technology, yet its adoption for regular communication was stagnant until the mass migration to remote work made it the de facto platform for many professional and personal encounters. This rapid, forced migration presents an unrivalled opportunity to probe different features of video conferencing systems. We capture perspectives at the height of the 2021 COVID-19 pandemic. The results present in this paper outline a more nuanced view of self-view, highlighting its advantages and disadvantages and presenting suggestions for improvement. In 2021, participants used self-view as a tool for verifying positioning, background, and facial expressions during meetings. While past research indicates that self-view may be disruptive and inspire feelings of self-consciousness, this can be balanced by the reassurance that arises from being able to monitor one's self-presentation. The discussion demonstrates the cultural shift caused by the wide-spread adaption of VC technology and illustrates the fluidity of societal norms on the format of the interaction.

## 1 INTRODUCTION

Alongside social distancing, mask wearing, and quarantining - video conferencing became the 'new normal' since 2020. Due to the stressors of the global COVID-19 pandemic, the modern human has been working from home. The resulting mass adoption of remote work has presented a unique opportunity to study the culture of workplace transitions in various forms. In response, researchers have been active in exploring both the benefits and the shortcomings of technologies, particularly video conferencing [3, 13, 17, 19, 29, 31, 33, 37, 38, 47, 48, 52]. These studies have examined the effects of factors such as image size, proximity to the camera, purpose of the call, and text transcripts on communication in Video Calls (VCs).

People have regularly used video conferencing as the de facto standard for collaboration since the widespread lockdowns in response to COVID-19. Over 25 years ago, Hollan and Stornetta's seminal paper [26] on video conferencing highlighted how the unique spaces created by collaborative technologies can be leveraged to present different – as opposed to simply better or worse – ways of working and how experiences with these technologies create new – as opposed to simply better or worse – ways of communicating. Examples of the benefits of video meetings include greater accessibility and savings in travel cost and time [22]. Researchers have, however, highlighted some of the shortcomings of video conferencing [40, 41, 45, 61].

Unlike traditional communication in vivo, VC also presents features specific to digital platforms including the self-view window. Factors such as self-consciousness, the mirror effect, and movement coordination have all been discussed to highlight why seeing yourself in remote meetings is undoubtedly problematic. However, most of these claims on the effects of the self-view are, to our reading, theoretical, based upon results inferred from other domains [25, 42]. As a result, we ask:

---

*e-mail: anon

**RQ1**: How is the self-view feature used during video calls?

**RQ2**: What are the benefits and drawbacks of the self-view feature in video calls?

To explore these questions, this paper presents the results of 17 data points consisting of semi-structured interviews in 2021. In the interviews, participants were asked to discuss their lived-experiences regarding the transition to Work From Home (WFH), VC experiences, and approaches to using features presented only in the digital form, e.g. the self-view window. These results are interesting in two ways. First, from the perspective of ecological validity, one benefit to exploring these questions during the pandemic is that the meetings people engage in are real world meetings with a specific goal (rather than constructed for a specific study). As a result, there is nothing artificial about the meetings themselves, permitting a more ecologically sound basis for data collection. Second, while meetings for teams have become virtual by default during the pandemic, it is also highly likely that – due to cost and or environmental concerns – many of the successes of online meetings may present a more durable change in the way people work and collaborate. Hybrid, remote, and distant work has become more normalized and, correspondingly, the understandings arrived at through the pandemic are highly likely to be durable components of work in the post-pandemic new normal. Overall, our results highlight both the advantages and shortcomings of the self-view during VC meetings in 2021. We find that self-directed attention was helpful in monitoring and improving one's self-presentation. Although excessive use of the self-view feature can cause anxiety and fatigue for some users, understanding and navigating through the social etiquette of camera use can help to alleviate these effects.

## 2 RELATED WORK

The progression of communications technologies have led us to closer imitations of in-person interactions, however, the missing sense of social presence is the prevailing limitation. New features of VCs, particularly the self-view, present opportunities to further explore this issue. In this section, we explore research on optimal communication, alternatives to in-person communication, the shortcoming of video calling, and the effect of mirrors/self-view on behaviour.

### 2.1 How we typically communicate best

There have been drastic changes in how we communicate with one another in the past few decades, with the introduction of telecommunications and online platforms [16, 54]. Despite the convenience of these newer methods, the literature has long held that video interactions are not functional alternatives or replacements for in-person interactions [16]. Past research also tends to agree with this theory. For instance, online therapy, Skype interviews, and telehealth have been considered as fallback options, and although beneficial for some, are a long way from being accepted as the preferred method for most [7, 27, 39, 54].

What is different about in-person communication? Face-to-face conversation simply offers more information. For example, non-verbal behaviour, being difficult to regulate and conceal, commu-

nicates valuable information in a face-to-face conversation and is important in building trust [11]. A more recent study by Brincke and Weisbuch [56] showed that the consistency of verbal and nonverbal behaviour results in communication coherency, which influences the perceived truthfulness of the message. In-person conversation also allows for the ability to better respond to each other and the information presented as shown by a study comparing the performance of drivers conversing on the phone versus those conversing with a passenger in the car [20]. If both individuals are present in the same environment and aware of the same stimuli, they can better regulate and modify the conversation because they are both aware of the demands of the situation.

The unique characteristics of in-person communication called for defining the idea of social presence. Social presence was first defined by Short, Williams, and Christie as "the degree of salience of the other person in the communication and the consequent salience of the interpersonal relationships" [51]. Promoting social presence helps to achieve a sense of cohesion and community in interpersonal relationships [59] and is an important factor to consider in digital communication [32].

## 2.2 Alternatives to in-person communication

According to Short, William, and Christie, the degree of social presence depends on the characteristics of the medium itself [51]. This section will briefly review the ways that different platforms afford social presence.

In texting, given the lack of verbal and non-verbal cues, it can be particularly difficult to convey a sense of social presence or togetherness [53]. When communicating with strangers through text, for instance in customer service, social presence can be promoted by providing identifying information, such as a name [53]. On the other hand, when texting among friends and family, it is common to imagine each other's voices and expressions as to create the experience of being there in-person [15]. Emojis are another addition that can help to convey emotions, which help to strengthen social relationships [43].

VCs seem like the best option for imitating in-person interactions. A medium that offers both video and audio feedback would presumably provide the strongest sense of social presence. However, when commercially introduced in 1964, the 'Picturephone' lost steam due to poor quality and general dislike of the idea and inconvenience at the time [58]. In the late 90's, inconvenience, concerns with privacy, and lack of training or familiarity continued to be roadblocks to widespread use of VCs [58]. In the context of the COVID-19 pandemic, the use of VCs is more prevalent than ever, considering the need for socially-distanced communication [21]. Moreover, media richness does not necessarily improve communication [18]. In a study comparing collaboration through audio versus VCs, the level of social presence and engagement was more strongly influenced by group cohesion in pre-established groups, than by the differences in the medium of communication [60]. Therefore, more information does not always warrant better communication. The next section explores the shortcomings of VCs that contribute to these feelings.

## 2.3 Shortcomings of VC

Although visual feedback is provided in VCs, not all aspects of non-verbal communication registers. For instance, VCs cannot provide eye contact and typically the gaze is slightly downward when looking at the video feed of others in the call [5]. Mutual gaze is important for communicating understanding, attention, and turn-taking cues [36]. In an experiment where a video conferencing system provided the possibility of eye contact, the participant's behaviours (i.e. direction and duration of gaze) were similar to in-person situations, implying the increased sense of social presence [36].

A feature that may contribute to the distant feeling of VCs is the rigidness and decontextualized nature of the 2D images presented.

One way to overcome this issue is to place the video feed of participants in a 3D setting (e.g. around a virtual table) [24]. This increased social presence from the original 2D condition, although not as high as the face-to-face condition [24].

An intriguing feature of video conferencing is the self-view option, which can result in a feeling of awkwardness. For example, in a workplace setting, introverted individuals were less likely to use video conferencing tools available. Some concerns of these individuals were the feeling of self-consciousness, increased self-attention, and the discomfort of being watched [58]. Another concern is communication apprehension, defined as the anxiety of actual or anticipated communication with others. This anxiety is exaggerated due to the inefficacy of VCs and increased self-attention, which results in avoidance or withdrawal behaviours [6]. VC users become more focused on how others perceive them due to the presence of the camera and self-view window [34]. Interestingly, being on camera and not receiving feedback through the self-view is also associated with higher social anxiety [30]. In the workplace, the constant monitoring through the self-view has been found to be a contributor to fatigue [50]. With so many factors to consider with VC culture and the use of the camera, this paper seeks to understand how VC users have fared in the context of collective WFH environments.

## 2.4 Understanding the self-view

It is known that the mere presence of mirrors impacts our behaviour in many ways. For instance, the presence of mirrors influence which foods we purchase at the supermarket, whether we deviate from normative behaviour, or how we respond to moral dilemmas [12, 42, 49]. The cause for these changes in behaviour is due to the self-focusing effects of mirrors resulting in an increased attention to one's own feelings and actions [10, 46]. The impact of mirrors can be varied based on individual dispositions. For example, performance is inhibited for individuals with lower self-esteem when the task is completed in front of a mirror [46].

It is important to consider that the self-view in VCs is not exactly a mirror. Varied size or shape of image, a reversed image, camera quality, and lighting changes can present a slightly different version of self than the one seen in mirrors. The self-view does however show the user exactly how others will view them in a VC.

## 2.5 A new normal

Novelty effect is caused by greater initial interest in a new technology, which diminishes over time [44]. As discussed earlier, VC are not a new invention; however, their adoption as the primary form of communication during lock-downs is a novel method of use. In the 90's, papers on the earlier implementation of VC showed that users were initially attracted to VC technology, which may have motivated their use and impacted their heightened sense of awareness and discomfort [9, 55]. However, over time the novelty of the technology wore off. Even with increased familiarity, the mass adoption of VC took place decades later during the COVID-19 pandemic. Perhaps in the adoption of a new technology, both the novelty effects of the technology itself, and the novelty of the way that it is used need to be considered.

Now, VC use has become normalized, and even essential, and users recognize the benefits brought on by this mass adoption. Users recognize advantages of working remotely such as saving on travel and spending more time with family. The difficulties of using VC to facilitate all meetings have also become apparent, such as having to learn a new skill and virtually forming meaningful social connections [1, 4]. To cope with their unique challenges, users have collectively developed VC etiquette that is beginning to be explored in literature [28].

This paper aims to further explore not only how users have adapted to VC, but also how they have adapted features of VC, such as the self-view, to their needs. The gradual return to in-person

work and variations of hybrid work settings grants users the flexibility to choose their ideal workspace. This presents an opportunity for understanding the lessons learned and gaining insight on the future of WFH.

## 3 METHODOLOGY

Given the impact of the COVID-19 pandemic, our research captures data during the height of the lock downs where VC remained one of the few ways to connect with other people. Within our work, we aim to understand how features of VC, especially those not present in vivo communication (i.e., the self-view in VCs, muting, and real-time video modifications including virtual backgrounds) influence the user's experience and impacts our culture. To explore the research questions, this paper features the compiled results of a qualitative, semi-structured interview protocol. After informed consent was obtained, participants were asked to share experiences that were most relevant to them in the framework of the core questions posed in an online recorded interview. Data was collected and analyzed using a grounded theory approach [8].

### 3.1 Participants

In total, we have 17 participants aged 21-50, 9 of whom identified as males and 8 as female. Participants identified as undergraduate, graduate, and PhD students all of whom participated in research or in co-op experiences. We focused on students involved in research teams due to their experiences in using VC for learning, working, and socializing. Of the 17 participants, 10 were studying or researching in the field of computer science and engineering. The remaining 5 participants were studying in the fields of biology, physics, or psychology.

### 3.2 Interview Protocol

The interviews were exploratory and semi-structured allowing for open-ended responses and follow-up inquiries. The core questions can be grouped into the following categories:

- comparing life before and after the COVID-19 pandemic

- differences between in-person and VC interactions

- the participant's routine for preparing themselves and their space for VCs

- how participants approached the use of the self-view feature

- approaches to online self-presentation and professionalism

Interviews were conducted one-on-one in Microsoft Teams by the first author. The video and audio of the interviews were recorded and transcribed with consent from the participant using otter.ai software. The video component of the interviews was deleted. The preparation of the data resulted in anonymous transcripts excluding introductions, questions, and any identifying information.

### 3.3 Interview Analysis

Interview transcripts were analyzed using grounded theory methodology [8]. Grounded theory is an inductive approach to data analysis which involves building a theoretical model as incoming data is collected. Researchers analyze the data by identifying reoccurring codes and later broader categories which continue to be supported as incoming data is analyzed. The grounded approach allows for iteration between the data and the developing theories; this approach allows for flexibility but demands the continued support of developed ideas. NVivo was used to code the data with the tags. Based on the coded and clustered interview responses, the codes were then further clustered into major themes. The organization of the recurring ideas, tags, and major themes is displayed in 1. In line with grounded theory, we collected data with the aim of reaching saturation. Saturation refers to the exhausting of new ideas emerging from data collection [2].

## 4 RESULTS

From the data collected and post-analysis, we present our main findings: in-meeting behaviour is moderated by the unspoken **social contract** and **personal expectations**. While conversing we wear a **social mask** to meet these expectations. The limitations of the VC medium further necessitate the use of the social mask. Finally, we find that the **self-view** available during VCs causes further differentiation from comparable in-person interactions.

### 4.1 Social Contract

There are unwritten, and oftentimes ambiguous, rules that dictate how we present ourselves and participate in a group VC. The goal is to seamlessly fit in while navigating through the technical difficulties and the awkwardness that may result. This section is about creating and understanding these rules.

#### 4.1.1 Social Fit

In an academic research setting, the choices regarding the camera are left to the participants. The process of deciding when to turn the camera on and for how long can be quite elaborate.

> " I remember one time someone had their camera on. But I didn't want to turn my camera on, because I didn't expect that it was going to be video. But they turned it on. And it was like oh, I feel like it'd be rude not to turn it on. So I had to." - P1

Although participants generally preferred meetings where cameras were turned on due to higher levels of engagement and social presence, they expressed discomfort with having to constantly present themselves visually to the group. Ultimately, they would conform with the rest of the group on their camera decisions and turn their camera on as a way of creating a more welcoming and engaging environment for others. The quotes below demonstrate an example of the decision making process:

> "If no one else has their camera on, I'm definitely not going to. If I'm in a very large seminar and only the speaker has their camera on, then sometimes I turn my camera on to make them feel less lonely because some people have said that they feel lonely as a presenter in seminars. If it's a really long meeting where people are taking turns to speak, I will keep my camera on until my speaking turn and then turn it off afterwards." - P13

The decision to turn the camera on or off mattered to the participants because it was one of the ways in which they made their presence known and in turn felt a sense of belonging. Furthermore, they helped other team members feel the same sense of belonging by turning the camera on when appropriate.

> "When they turn their video off, I felt like I'm just talking to a bunch of text. Whereas when they put their video on, I can see their facial expressions. And it just made me feel more connected with them." - P10

Overall, participants felt more engaged in meetings where most group members had their camera on, and they appreciated the option to turn their cameras off at times when they felt fatigued.

> "[Putting my head down] looks bad on a video call, so if I feel like I'm really tempted to do that, then I would definitely turn my camera off." - P13

| Recurring Codes | Category | Major Theme |
|---|---|---|
| Managing one's own participation; conformity; empathy - social interpretation (reading the room); fitting in | Social Fit | Social Contract |
| Social navigation of technical concerns; conversation flow; silence; interruptions | Group Communication | |
| Glancing, watching, or avoiding the self view | Self-monitoring Actions | Personal Expectations |
| Professional demeanor and appearance; checking gestures for communication; checking and preparing setup | Appearance | |
| Checking expressions; exaggerating expressions (for clear communication), seeming attentive/interested/focused | Emotional Communication | |
| Assumptions, uncertainty, distrust of others; intentions/authenticity; discomfort with sharing content | Distrust | The Social Mask |
| Content of the meeting; types of conversation; changes in the expectations | Purpose of Meeting | |
| Lagging; lack of body language; lack of eye contact | Technical Issues | |
| Self-reflection, critical thoughts, worry about narcissism | Anxiety | Mixed Feelings |
| Video call fatigue and fatigue of checking the self-view | Fatigue | |
| Bored, distracted, interest dependent on meeting type | Distraction and Disinterest | |
| Comparing internal expectations (feelings or appearance) with self-view output | Reassurance | |

Figure 1: Clustering of recurring codes, categories, and major themes

### 4.1.2 Group Communication

Participating in a group conversation in VCs is hindered by our impaired ability to read the room and the technical difficulties. Transition sentences and audio cues are the most helpful as opposed to the camera.

> "...when I'm definitely not talking, mute. I should not leave it on unless I'm pretty sure I have something to say. Because, again, I cannot read the room. I cannot see when I can jump into the conversation and also as a way to prevent annoying background sounds on my end from intruding on others." -P8

Sometimes, things are left unsaid because of how difficult it can be to interject without interrupting. Overall, communication is more reserved and concise. Facial expressions or emoji reactions can help to make up for the lack of communication.

> "I think I participate a lot less because of the lag. And I feel like if I talk, there's more of a chance I would accidentally be talking over someone. Whereas in in-person meetings I'm like, I put myself out there a lot more. And I say a lot more ideas." -P13

> "Sometimes if I feel like I have not been able to get a word in at all. I , like, kind of, like, weird, like, squint my eyes or something just kind of feel like slight discomfort or something. And I hope that he will pick up on that. And in a similar way, I pick that up on other people, or if someone's kind of like, like staring off into the distance or something, I know that they're thinking about some things, I know that I should check it out with them, once this person stops speaking." - P16

Participants felt that they were responsible for maintaining the flow of conversation despite the technical difficulties. The fear of

interrupting resulted in a hesitance in participation. They had to dedicate extra effort into deciding when to speak and how much to contribute based on the participation of others.

### 4.2 Personal Expectations

Conforming to the social contract was achieved by presenting the most socially competent self when in front of the camera, or in other words putting on a 'social mask'. VCs grant more control and awareness to one's appearance and emotional communication. The self-view granted users the ability to self-evaluate to a higher extent than in in-person interactions.

#### 4.2.1 Self-Monitoring Actions

Participants inevitably check the self-view window. It is a unique feature in communication since one is able to view themselves precisely as seen by others. Therefore, participants used this feature to evaluate how well they abide by the social contract and if they meet their own expectations of self-presentation.

> "I noticed like during my meetings. If I - if I am on camera, I'll look at my self view a lot. constantly checking like you're not like yawning or you know, scratching your head or anything like ... how other people will see you? "- P9

In the above quote the participant walks through their process for self-monitoring. Self-monitoring behaviour was used to ensure participant appears engaged and professional. Appearance during VCs was discussed specifically during lab meetings and interviews.

#### 4.2.2 Appearance

The goal communicated by the participants is to present a professional image of oneself to the group. The simplicity of the background contributed the most to the look of professionalism; specifically, avoiding the display of clutter, personal items, and distractions

in the background. For most participants this involved positioning the camera in a way that showed as little of their personal space as possible. If participants had more resources and time available to them, they expressed an interest in personalizing their backgrounds to show their interests or affiliations.

> "So it's always a conscious effort to make sure that not much of my personal spaces visible to the others. And that kind of hides a person's individuality. But when you go to an office space, when I go to my advisor's office, I can see photos of her children ...the way she organizes the office, the way she the places her plants on the desk, that kind of gives an individuality a sense of how the person is" - P4

Participants treated their backgrounds as curated snapshots of their lives. Given that what is presented is in the form of a live video feed, the self-view window comes into play for monitoring this snapshot and any changes.

> "Because the moment you turn on the video, you're like consciously like staring and you cannot look around...And it's at times I've seen people just turning their video off and then drinking water, something as simple as drinking water, which is something you would do in an in person conversation very casually." - P4

Participants were wary of distractions such as pets or people, or simple actions such as drinking some water; this level of self-monitoring and cautiousness regarding presented appearance differs greatly from in-person meetings where individuals do not excuse themselves to drink water. In contrast, in-person presenters may use drinking water as a method to pace themselves.

### 4.2.3 Emotional Communication

An important part of professionalism is communicating clearly – verbally and non-verbally. Perceived differences from in-person communication may encourage greater expression of non-verbal cues. Concerns discussed by participants indicate there is a higher emphasis on emotional communication.

> "I don't know if it's maybe because like it's because of the selfie camera or because I feel like with video chat is more restrictive and being able to show your, like body language, but like sometimes I'll monitor it just to see if like - like I can like show - I'm making sure I'm showing like a positive effects like a more exaggerated like happy face or like making sure I - like I'm paying attention" - P14

Participants noticed that they emphasized their interest, engagement, and attention by nodding and smiling in a more pronounced manner than usual. They exaggerated said actions due to their lack of faith in their efforts in being noticed or sufficiently understood in the VC medium.

### 4.3 the Social Mask

The perceived shortcomings of the video conferencing platform caused changes in behaviour. The limitations of video conferencing as a medium were clearly expressed by participants. our results suggest limitations of this medium necessitate and encourage the use of the social mask.

### 4.3.1 Distrust

We found two factors resulted in a feeling of distrust in participants: concerns about privacy and the candidness of others.

> "Even though this is not recording, then I get a sense that, you know, everything online will be saved. You know, like, there's always someone that can access the data"-P9

Participants felt reluctant to share information due to the fact that online interactions permanent and lasting, even when meetings were not recorded. There was also the worry of their image or words being taken out of context at a later time.

> "Sometimes you forget that you're in a call, especially if someone is presenting something boring. And you start to behave weirdly. And then someone takes that screenshot, and it's shared all over the groups." - P5

This quote illustrates that it can sometimes be difficult to maintain this social mask, especially when in a comfortable space such as one's home. Presenting a professional image of oneself is can be more taxing in the WFH video call set-up. Overall, participants felt that they had to be on guard by putting on and maintaining their social mask.

### 4.3.2 Purpose of VC Meetings

VCs are very structured unlike more free-flowing in-person meetings, yet the experience feels less professional due to each person's surrounding home environment. This unusual combination is ideal for structured meetings such as presentations and job interviews, which participants found were less stressful in a VC format. Participants expressed disappointment for more spontaneous meetings, for instance socializing or informal lab meetings.

> "I think in person, there's more room for just regular small talk than there is on zoom meetings, because it feels like, like you scheduled time out of your day for this, so it feels like it needs to be more packed with important things. So I'm less likely to kind of just share some random, like fun thing that I did over the weekend or something." -P6

The video conferencing format favoured a more reserved approach to contributing to meetings. Participants presented a more reserved, and therefore professional, version of themselves with the help of the social mask.

### 4.3.3 Technical Issues

There are plenty of technical issues in video conferencing that limit communication such as the lag, lack of eye contact, or lack of body gestures, etc. The next quote demonstrates how participants compensate for these shortcomings using the social mask.

> "Because now that I think about it, because of the loss of immediate verbal cues, I find sometimes I will explain explicitly, try to have certain facial gestures. ... I feel like, you know, I should do my part, and like, at least provide some visual feedback to show that I'm listening." - P8

In the above quote, P8 discusses the use of overt facial gestures to add information and aid comprehension during video meetings. This extra effort (i.e. exaggerating facial expressions) is how participants attempt to compensate for technical issues using the previously discussed social mask. Participants used the social mask to make-up for information that they felt was lacking in communication.

### 4.4 Mixed Feelings

Adhering to the social and personal expectations through the use of the social mask, although reassuring to some extent, results in feelings of anxiety, fatigue, and distraction.

### 4.4.1 Anxiety

Seeing our own image prompts us to evaluate our appearance and behaviour. For some, the critical thoughts resulted in self-consciousness and lower self-esteem. Meetings were more comfortable when participants had the option to disengage to some extent by turning the camera off. Participants also found it beneficial to be able to conceal parts of their reactions or emotions.

> "I was so uncomfortable in that meeting, because it's for senior management, and I'm just a co-op student. So well, while other people are turning their camera on, I turned mine off. ... I usually use like turning my camera off as a way to hide my anxiety hide my face, or especially because I don't want people to look at how like nerve racking I feel." -P10

### 4.4.2 Fatigue

Fatigue results from constantly appearing to pay attention, back-to-back meetings, or the physically static nature of calls.

> " It's just too much, you know, like, one time I missed a meeting, and I had to call them in a parking lot ... sometimes I do miss the meetings because I - I can't keep up." - P12

The constant monitoring of self-view feature causes fatigue for some participants, although it can be avoided by taking breaks and turning the camera off when possible, or simply being less stringent with the monitoring.

> " There's been countless times where I've been like, very, very tired on a call. And so I'll turn my camera off, and I'll kind of like, zoned out for a while. And that's something you wouldn't be able to do in an in-person meeting. And so sometimes I tend to drop the ball, because I can." - P16

### 4.4.3 Distraction and Disinterest

Our results demonstrate that participants felt it was inevitable to be disengaged from the content being presented in calls. The reason for this was that they were not fully immersed in the meeting and had distractions easily accessible to them, within and outside of the call itself.

> "You're nervous, probably you have a stress and you see yourself in your like physical place. But when it's a virtual so you can just if your webcam off so easily, you can be distracted by anything. ...Because you can open other tab you can look at other things while your camera if off and maybe you are not noticed that your eyes movement is obvious. " - P3

While the camera is turned on, the self-view window can be distracting, which is why some choose to avoid it when it is time to contribute to the meeting. There are many opportunities to turn the camera off (a break from the social mask), and these moments of resting can be a distraction as well. Therefore, participants believed that they would be distracted regardless of their camera choices.

### 4.4.4 Reassurance

Participants found that focusing on the self-view amplified certain feelings. At times when participants felt self-conscious, the self-directed attention affirmed this feeling and they preferred to turn their camera off. However, in moments when participants appreciated their appearance or demeanour, watching the self-view amplified their positive feelings and enhanced their confidence.

> "If I'm pressured to turn it on, and I don't feel great about myself that day, for whatever reason, it could be both like, emotionally, I'm not feeling well, or like, I just don't like my appearance, that will make me feel less confident. But if I am enjoying my appearance, I'm feeling confident that day emotionally and whatnot, then like, it tends to enhance myself, because I see - I perceive myself in a positive light. " - P16

Moreover, the self-view was helpful in detecting discrepancies between how the participant envisioned their self-presentation, and how they actually presented themselves.

> "I'm in a meeting, for me a meeting, and things got tough, or emotional, I like to check my self-view mirror to make sure I don't appear hostile. ... I tend to frown a lot, and I tend to look up a lot. And I know that can sometimes feel off-putting for people. So I often use the self-view, you are to check my facial expression, to make sure I do not appear hostile. " -P10

Overall, users found it challenging to get used to the self-view feature, however, they considered it to be a necessary and even helpful feature in presenting a positive and professional image of themselves, when used in moderation.

## 5 DISCUSSION

The rapid switch to video conferencing as the main medium for both personal and professional communication has provided researchers the opportunity to gather data. Due to the unprecedented societal shutdowns due to COVID-19, our paper investigates the lived experiences of individuals participating in the WFH paradigm. To conduct this study, we interviewed undergraduate and graduate students about the use and self-perceived effects of the self-view feature in their VC experiences. The data illustrates the effects of blending of the home and office environments prompts the exploration of changes in self-presentation and expectations of professionalism.

### 5.1 Summary of results

We begin by revisiting our research question: *How does the self-view feature in VCs impact user experience?* The results of our research demonstrated that the participants in a VC meeting strove to meet mutual expectations regarding their manner of communication and appearance. The goal of this ever-changing social contract seems to be to maintain a level of professionalism after the transition to a work-from-home routine. Participants knowingly behave differently in VCs and present a more polished persona of themselves, which we name the social mask. Our data indicates that the use of the social mask is facilitated by monitoring live feedback from the self-view window. Participants found the chance to monitor and correct their appearance or manner of communication to be a useful feature. In particular, the self-view provided assurance to participants in moments of uncertainty about their expressions, appearance, or changes in the background. Monitoring the self-view was especially important in structured VC meetings such as interviews where the first impression matters.

### 5.2 Adapting to the Current Video Conferencing Culture

#### 5.2.1 Social Contract

Our results illustrate that there was a common understanding between participants of a VC which helped to establish their social presence in a team. For instance, choosing to turn the camera on is a statement in itself and it can be used to show one's interest and engagement. The timing of participation in a way that complements the flow of conversation and avoids disruption is likewise important. As a recommendation, at times when the rules are ambiguous and

contributing is difficult, the emoji reactions or chat can serve as fallback options. By adhering to the social contract, we present a more professional version of ourselves.

### 5.2.2 Self-Presentation

The appearance of professionalism has been extended to include backgrounds presented in VCs. The self-view window plays a role in managing this aspect of self-presentation. While the background set-up allows for new ways to express oneself, the sudden switch to the VC format has not given everyone the chance to prepare. It may be helpful to accept the changes in the expectations for a professional appearance to be more reflective of the surrounding environment of users. Another recommendation is to give users the autonomy to use their cameras when they feel comfortable. Since the self-view window is not a static image, it requires closer monitoring for changes. Given the sometimes unpredictable nature of a work environment at home, it would be convenient to have the option of turning the camera off to avoid causing a distraction. Ironically, the effort to constantly self-monitor one's behaviour depletes some of the resources needed for effective self-presentation [57]. Therefore, with individual differences in mind, it would be ideal if users could decide for themselves when they can afford to dedicate some attention to their self-presentation and when they should take a break by turning their camera off.

### 5.2.3 Maintaining the Social Mask

In recent literature, the self-view feature of VCs has been found to increase self-awareness, and consequently shape our perception of self, as informed by the looking glass self theory. Moreover, the various VC interactions are theorized to benefit from different amounts of self-directed attention. For instance, in online dating, the increased self-awareness may encourage participants to adhere to social norms, whereas increased self-awareness may be unnecessary in group meetings that depend on the productivity of participants [34]. In another study, participants performed group tasks more effectively in the absence of the self-view [23]. More generally, turning the camera on was linked with fatigue for employees due to the worries of self-presentation [50].

In our study, participants confirm the increased level of self-awareness through self-monitoring. The benefits of this were the more sociable and professional self-presentation both in terms of appearance and emotional communication. Despite the focus on managing self-presentation, in our findings, the self-view was not the main cause of VC fatigue. Factors such as scheduling difficulties and back-to-back meetings were also taxing.

Data reveals unspoken rules set about turning the camera on or off. Turning the camera on was not an obligation for an evaluative purpose, instead it was a way to show an interest in being present in the group. The increased autonomy with the camera use had a positive outcome in that it allowed participants to take advantage of the self-view at their own leisure. The unwanted effects of self-directed attention in VCs can be avoided by simply turning the camera off, however this would negatively impact the engagement and social presence in meetings.

### 5.2.4 A New Era of Communication

Video conferencing technology has been available for about five decades and its widespread use had been predicted in the past. However, initially due to the cost, and later due to general reluctance and shortcomings, it has not been embraced as a replacement for in-person communication needs [14, 35]. Even with the elimination of most issues in quality, most participants, predictably did not value video conferencing interactions the same as in-person counterparts. Aside from the minor technical difficulties, the main reason for their reluctance was the missing sense of presence. Although a VC cannot fully imitate an in-person meeting, its unique qualities and features

distinguish it from any other form of communication. As theorized by Hollan and Stornetta (1992) [26], it seems to be more worthwhile to focus on how one can take advantage of the attributes of a platform, rather than judge its merits on how closely it mirrors in-person communication.

The presence of the self-view window is one way that VCs differ from in-person interactions, and its design and method of use can be tailored to the user's advantage. The self-directing effects of the self-view can be either advantageous or disadvantageous depending on the context of use. While the self-view has recently been viewed as a cause of fatigue and a nuisance, our findings in the context of the pandemic and the unique WFH environment indicate that it can also be beneficial to the user's well-being.

Due to the novelty of the environment exacerbated by the abrupt change in our working culture due to COVID-19, participants faced higher levels of anxiety. Our results demonstrate that adaption does not stem from demonstrations and short-term use; instead true adaption requires effort and commitment from users as they continue to struggle with new features. As users become increasingly more comfortable with VC norms, participants were more in control of how they presented themselves. The social mask worn by participants is possibly an insulating and protective reaction to an uncontrollable situation and the novelty of the online environment. Habituation to the digital online collaborative space can result in personal and group comfort as we collectively desensitize.

With the popularity of hybrid work and study, going back to in-person as per the previous normal is no longer the aim; instead, we bring with us some of the benefits that we now see as viable in part due to large scale forced adaption. Hybrid options provide flexibility for users allowing for inclusive experiences in workplace culture. Flexibility is a key driver in the adaption of the hybrid workplace. Leveraging flexible and inclusive advantages allows for increased adoption; so, we can apply this prioritization to persuade users to adapt to innovation collaborative in technology. Flexibility in expectations with camera use and self-presentation can be one such way to create more inclusive and comfortable workplace culture. Demonstrated advantages of the WFH culture for inclusion and accessibility may act as the reward needed to motivate VC skill acquisition.

### 5.3 Contribution

The COVID-19 pandemic caused significant hardship, yet also provided an opportunity to study the previously unfathomable thought experiment: 'what if everyone switched to only online communication'. We find a more nuanced and ecologically valid perspective on the self-view. Given the high likelihood that at least some aspects of our changed work and social landscape will prove durable, our understanding of this nuance is a valuable window on current and future practice.

The focus of the paper relies not on collaborative actions, but instead looks to understand the individual as they enter a collaborative digital environment. The main contribution of the paper is that the individual's view on self-presentation and collaboration over the course of the pandemic allows for an intimate and personal look on computer-mediated collaborative environments.

### 5.4 Limitations and Future Work

A limitation common to interview protocols is the reliance on self-reported experiences; however, for this topic which focuses on personal experience, self-report allows for an exploratory approach to researching the self-view. Interviews allow for the exploration of implicit factors that participants are unaware of, discomforts, and self-reflection with the researchers. More robust measures e.g. using an eye tracker to determine the extent of use of the self-view in different meetings could be a next step for future studies to provide an empirical approach to understanding use of the self-view. Such a

study would likely involve deception and would need further ethical considerations.

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

## ACKNOWLEDGMENTS

Removed for Blind review.
