# OpenReview forum: "“All Eyes On Me”: User Perspective on Self-View in Work From Home Video Conferencing"
_graphicsinterface.org/Graphics_Interface/2023/Conference — Submitted to GI 2023_

### Official Review · Reviewer_Rneu · 2023-01-09
**use of video conferencing tools during work-from-home**

**Rating:** 5
**Confidence:** 3

**Review:**

This submission describes findings from an interview study that aims to explore challenges associated with video conferencing for work purposes considering the switch to remote work during the pandemic.  The paper is well-written and engaging, with an introduction that emphasizes the need to revisit findings from pre-pandemic work. I am not familiar with this literature and so cannot assess the novelty of this work with respect to other work that has examined remote work during the pandemic.

A major weakness of this submission is the participant pool, which consists solely of students, the majority of whom are in CS and engineering fields. The paper states that all students had experience with video conferencing as part of co-op or research.  One could argue that these experiences are training as opposed to work experience, particularly the research context, where students are typically trainees.  Without a description of the type of work participants were using video conference for, it is difficult to get a sense of whether the findings speak to general challenges with video conferencing tools or challenges specific to work contexts.  For example, are the participants reflecting on work contexts only, or are they also reflecting on their experiences in classes and other education or training scenarios? As it currently stands, there is a disconnect between the paper’s introduction and the data collection decisions.  I was very surprised not to see the characteristics of the participants discussed in detail in either the discussion or limitations sections. To contextualize the findings, it would be helpful to have a better understanding of the type of work, team size, what VC is used for, typically usage frequency, etc.

A second concern with the paper is that most of the findings have been reported in the literature (as acknowledged by the authors).  The paper indicates that a contribution of this work is that the data has been collected in a more ecologically valid manner than prior work, however, without contrasting methods with those used in prior work, the validity of this argument has not currently been established.

I can see potential for this paper, but in my opinion, more work is needed to surface the paper’s contributions.

---

### Official Review · Reviewer_kvx2 · 2023-01-13
**mismatched motivation, method and finding. result make sense but the difference from the literature isn't clear**

**Rating:** 4
**Confidence:** 5

**Review:**

The paper presents an interview study that looked into how 17 student participants experienced and strategized their use of video calls in 2021 during the pandemic. If the methods were done properly, the paper could potentially have empirical contributions on the practice of video-mediated communication in work and learning contexts. However, it appears that the paper did have a number of conceptual and methodological limits, rendering the contributions insubstantial.

Overall this paper is looking at an important and timely issue around the actual practices of video calls usage outside of lab and controlled study environments. This paper certainly is not the first study trying to look at this practice, but as the context of usage is constantly changing, it's certainly great to see a new report that's recently done in 2021. The qualitative findings around people's perceptions in terms of social contract, self monitoring, social mask and emotion (e.g., anxiety and reassurance) overall make good senses, though many of these seem to be not so surprising. The difference from what has been known and done in the literature was not explicit. So it's somehow difficult to assess the value of these qualitative observations.

With that said, the main issues with the paper are technical ones. The paper started with a framing that emphasizes on self-view, but the interview protocol appears to be broadly asking the interviewees their general experiences of video calls, and the interpretative results also scattered around all different aspects of video calls. The findings that directly speak to self-views are very limited (self-monitoring, reassurance etc.), and don't seem to be very surprising. There's also lack of clear implications for design iterations and applications.

I would suggest to tighten the review of the literature, focusing only on self-reviews, and make sure that the current findings are discussed with respect to what we already learned from the past. The direction remains important and interesting to the HCI community, but the method and results need show validity and value.

---

### Official Review · Reviewer_19RE · 2023-01-21
**Relatable survey of experiences with Zoom - but not novel enough for publication given recent prior work**

**Rating:** 4
**Confidence:** 5

**Review:**

The paper presents the results of an interview study with 17 people about their experiences using videoconferencing during a period where online meetings and lectures were common (due to Covid-19). The authors report several findings from the interviews related to social expectations in remote meetings and social presentation. The authors frame the paper around the idea of exploring the "self view" that is now ubiquitous in videoconferencing systems - but as described below, many of the study's results are more generally about people's experiences with videoconferencing rather than specifically about issues relating to the self view.

Although the experiences described in the paper are interesting and relatable, given that many people have been in similar settings during Covid-19, the research contribution of the paper is not clear. There are two main problems. First, the findings of the study are somewhat superficial, with single quotations from the interviews provided as the only example of a particular point that the authors are making. In addition, the paper's stated focus on self views is not actually maintained in the study: most of the findings presented are about videoconferencing in general, not the self view, and cover a wide range of issues including latency or reluctance to turn one's camera on. The results section of the paper is brief, and the three categories suggested by the authors as their organizing principles do not appear to be clearly defined and do not seem to add valuable insights to what we understand about videoconferencing or remote work.

Second, the paper does not put the study into the context of several studies and papers that have recently been published on videoconferencing and remote work during the Covid-19 pandemic. Some of these studies consider very similar issues to those that are covered by the submission's interview topics, and thus the novelty and originality of the submission's contribution is substantially weakened. For example, the following papers look both at the self view as well as general experiences of using videoconferencing systems:

Kuhn, K. M. (2022). The constant mirror: Self-view and attitudes to virtual meetings. Computers in Human Behavior, 128, 107110.

Gullo, N., & Walker, D. C. (2021). Increased videoconferencing after COVID-19 stay-at-home orders increased depression and anxiety but did not impact appearance satisfaction or binge eating. Computers in Human Behavior Reports, 3, 100080.

Wang, B., & Prester, J. (2022). The Performative and Interpretive Labour of Videoconferencing: Findings from a Literature Review on 'Zoom' Fatigue.

Vidolov, S. (2022). Uncovering the affective affordances of videoconference technologies. Information Technology & People.

Balogova, K., & Brumby, D. (2022, June). How Do You Zoom?: A Survey Study of How Users Configure Video-Conference Tools for Online Meetings. In 2022 Symposium on Human-Computer Interaction for Work (pp. 1-7).

These recent published results present similar findings and cover similar ground to the present submission, and without a clear understanding of how the submission's contribution differs from this prior work, it is difficult to see the paper as presenting a valuable and new set of insights about videoconferencing.

---

### Meta-Review · Area_Chair_Pex6 · 2023-01-21

**Recommendation:** 4
**Confidence:** 4

**Metareview:**

Reviewers all agree that the submission is on a highly relevant and interesting topic, with the potential for contributions.   Despite their enthusiasm for the direction, all reviewers felt that the paper is not ready for publication.  The reviews point to the following key issues:

- Reviewers noted framing and scoping issues in the introduction and findings.  The introduction focuses on “self view”, however, the interviews questions and findings cover a much broader spectrum of issues.  Consequently, the notion of “self view” feels underexplored.
- Distinctions with respect to related work (including very recent work) need to be strengthened.
- The paper would benefit from further reflection on the participant pool.